Genome-wide association study identifies novel type II diabetes risk loci in Jordan subpopulations

Dajani Rana rdajani@hu.edu.jo 1
Li Jin 2 3
Wei Zhi 4
March Michael E. 2
Xia Qianghua 3 5
Khader Yousef 6
Hakooz Nancy 7
Fatahallah Raja 8
El-Khateeb Mohammed 8
Arafat Ala 8
Saleh Tareq 1
Dajani Abdel Rahman 1
Al-Abbadi Zaid 1
Abdul Qader Mohamed 1
Shiyab Abdel Halim 9
Bateiha Anwar 6
Ajlouni Kamel 8
Hakonarson Hakon hakonarson@email.chop.edu 2 5 10
1 Department of Biology and Biotechnology, Hashemite University , Zarqa , Jordan
2 Center for Applied Genomics, The Children’s Hospital of Philadelphia , Philadelphia , PA , United States of America
3 Department of Cell Biology, Tianjin Medical University , Tianjin , China
4 Department of Computer Science, New Jersey Institute of Technology , Newark , NJ , United States of America
5 Divisions of Human Genetics, The Children’s Hospital of Philadelphia , Philadelphia , PA , United States of America
6 Department of Community Medicine, Public Health and Family Medicine, Faculty of Medicine, Jordan University for Science and Technology , Irbid , Jordan
7 Department of Biopharmaceutics and Clinical Pharmacy, School of Pharmacy, University of Jordan , Amman , Jordan
8 National Center for Diabetes, Endocrinology and Genetics , Amman , Jordan
9 Department of Anthropology, Yarmouk University , Irbid , Jordan
10 The Perelman School of Medicine, University of Pennsylvania , Philadelphia , PA , United States of America
Foti Daniela
Electronic publication date: 2017 Aug 17
Publication date: 2017
Volume: 5
Electronic Location ID: e3618
Received 2017 Apr 3; Accepted 2017 Jul 6
Copyright: ©2017 Dajani et al.
Copyright year: 2017
Copyright holder: Dajani et al.
License: This is an open access article distributed under the terms of the Creative Commons Attribution License, which permits unrestricted use, distribution, reproduction and adaptation in any medium and for any purpose provided that it is properly attributed. For attribution, the original author(s), title, publication source (PeerJ) and either DOI or URL of the article must be cited.
License URL: https://creativecommons.org/licenses/by/4.0/

Keywords: eQTL, Methylation, Meta-analysis, Type 2 diabetes, Genome-wide association study

Funding: King Hussein Center for Cancer and Biotechnology Higher Council for Science and Technology, Amman, Jordan The Children’s Hospital of Philadelphia to the Center for Applied Genomics This work was supported by the King Hussein Center for Cancer and Biotechnology and the Higher Council for Science and Technology, Amman, Jordan, and by an Institutional Development Award from The Children’s Hospital of Philadelphia to the Center for Applied Genomics. There was no additional external funding received for this study. The funders had no role in study design, data collection and analysis, decision to publish, or preparation of the manuscript.

==============================
The prevalence of Type II Diabetes (T2D) has been increasing and has become a disease of significant public health burden in Jordan. None of the previous genome-wide association studies (GWAS) have specifically investigated the Middle East populations. The Circassian and Chechen communities in Jordan represent unique populations that are genetically distinct from the Arab population and other populations in the Caucasus. Prevalence of T2D is very high in both the Circassian and Chechen communities in Jordan despite low obesity prevalence. We conducted GWAS on T2D in these two populations and further performed meta-analysis of the results. We identified a novel T2D locus at chr20p12.2 at genome-wide significance (rs6134031, P = 1.12 × 10−8) and we replicated the results in the Wellcome Trust Case Control Consortium (WTCCC) dataset. Another locus at chr12q24.31 is associated with T2D at suggestive significance level (top SNP rs4758690, P = 4.20 × 10−5) and it is a robust eQTL for the gene, MLXIP (P = 1.10 × 10−14), and is significantly associated with methylation level in MLXIP, the functions of which involves cellular glucose response. Therefore, in this first GWAS of T2D in Jordan subpopulations, we identified novel and unique susceptibility loci which may help inform the genetic underpinnings of T2D in other populations.

Introduction

Diabetes is among the most common non-communicable diseases globally. It has been estimated that there are currently about 194 million people at the age of 20 to 79 with diabetes worldwide and that this number will further increase to 333 million by 2025 (Wild et al., 2004). Diabetes is the fifth main cause of death in Jordan, afflicting 16 percent of Jordanian adult citizens; another 23.8 percent of adults in Jordan are also on the brink of becoming diabetics according to a study from 2007 by the Heart and Capillary Disease Prevention directorate (HCDP) of the Ministry of Health in Jordan; and the rate of diabetes prevalence in Jordan is 30.5 percent among both children and adults (Ajlouni et al., 2008). Thus, diabetes presents a significant public health burden to the Jordan community. Type II Diabetes (T2D) is the major type of diabetes, which accounts for 95% percent of all diabetes cases worldwide.

Despite extensive research efforts for more than a decade and some notable successes, much of the genetic basis of common human diseases remains unresolved (Hirschhorn & Daly, 2005). The genome-wide association study (GWAS) has been a powerful approach for identifying novel susceptibility loci for complex diseases (Barrett & Cardon, 2006; Pe’er et al., 2006), such as T2D. To date, more than 80 T2D susceptibility loci have been uncovered by GWAS. However, the heritability attributed to these loci remains as low as just 10% (Imamura et al., 2016). In addition, these studies have mostly focused on populations of European ancestry and East Asians, with a few studies on South Asians and Mexicans. The genetic determinants of T2D in Middle East populations have not been extensively studied by GWAS and limited evidence suggested that at least some of the reported T2D loci showed differential associations in different populations in the Middle East (Mtiraoui et al., 2012). It has also been reported that the presentation of T2D is different between Middle East immigrants and European patients (Glans et al., 2008), implying some different genetic basis between populations. Given the prevalence of the disease in the region, more research is warranted to understand the genetic basis of T2D specific to given Middle Eastern populations.

The Circassians and the Chechens are two ethnic populations of ancient descent in Jordan, both of which are the largest indigenous nationalities of the North Caucasus (Barbujani, Nasidze & Whitehead, 1994; Bulayeva, 2006; Nasidze et al., 2001). These two populations are descendants of a single ancient origin with later divisions along linguistic and geographic borders (Nasidze et al., 2004; Nasidze et al., 2001). After immigrating to Jordan 140 years ago, Circassians and Chechens in Jordan are endogamous and have managed to keep their separate sense of identity and ethnicity during the last one hundred years in Jordan (Kailani, 2002). Previous analysis of classical genetic markers such as blood groups and serum proteins have also shown statistical significant genetic diversity in the Caucasus (Barbujani, Nasidze & Whitehead, 1994; Barbujani et al., 1994), which has been further confirmed by mitochondrial DNA and Y chromosome analysis (Nasidze et al., 2004; Nasidze et al., 2001). While a T2D GWAS has been conducted in the Lebanese population (Ghassibe-Sabbagh et al., 2014), the Lebanese are Arab in origin; Circassians and Chechans are a separate, non-Arab ethnic group. These are clearly different populations, with different ancestries. The Circassian and Chechen communities may provide us an opportunity to study a genetically unique population and compare genetic basis for complex human diseases between different populations.

T2D has become an alarming public health issue in Jordan. Epidemiology studies showed that the prevalence of impaired fasting glycemia is 18.5% and 14.6% and prevalence of diabetes is 9.6% and 10.1% for Circassians and Chechens, respectively (Dajani et al., 2012). In view of the very high incidence of T2D in Jordan and the genetic distinctness of Circassian and Chechan populations, we performed a GWAS to search for genetic factors contributing to T2D in these two populations and compared the results with European population.

Materials & Methods

Ethics statement

The study has been approved by the institutional review board committee at the National Center for Diabetes, Endocrinology and Genetics of Jordan (approval number: 457/9.MS). The written informed consent was given by all participants.

Study subjects and sample collection

A random sample of N = 144 from the Chechen population in Jordan and a random sample of N = 140 from the Circassian population in Jordan were recruited to participate in the study. Each participant in the study filled out a survey that included pedigree information. The identities of parents, grandparents, and great-grandparents (both maternally and paternally) were reported in the survey and any individual with non-Chechen heritage for even one person in his/her pedigree was excluded for the Chechen subpopulation; the same identity confirmation was conducted for the Circassian subpopulation.

A subject was defined as affected by diabetes mellitus if this diagnosis is known to the patient or, according to the ADA definitions, if fasting serum glucose is 7 mmol/L (126 mg/dl) or more. Impaired fasting glucose was defined as a fasting serum glucose level of ≥6.1 mmol/L (100 mg/dl) but <7 mmol/L. The glycemic control was assessed using HbA1c. Patients with previously diagnosed diabetes who had HbA1c >7% were defined as having ‘unsatisfactory’ glycemic control.

Sample collection

A total of 9 ml of whole blood was drawn in EDTA tubes from the subjects by vacutainer system. Genomic DNA was isolated from whole blood sample using the phenol-chloroform protocol.

Genotyping and quality control

We performed high-throughput, genome-wide SNP genotyping, using the InfiniumII OMNI-Express BeadChip technology (Illumina), at the Center for Applied Genomics (CAG) at the Children’s Hospital of Philadelphia (CHOP), USA. Sample quality control (QC) was performed based on the following measures: sample call rate, overall heterozygosity, relatedness testing and other metrics. Samples were excluded from analysis for SNP call rate <95%, heterozygosity beyond five standard deviation of the mean. One sample from each pair of duplicated or cryptic related samples was removed. For each pair of duplicate or related samples the sample with the highest SNP call rate was kept in the dataset. In the SNP-based QC, SNPs with a call rate <95%, minor allele frequency <1% or showing significant deviation from Hardy-Weinberg-Equilibrium (HWE test P-value <10−4) in the controls were removed. All QC steps were carried out using the software package PLINK (Purcell et al., 2007).

Principal component analysis (PCA)

PCA was conducted to confirm ethnic identity and to generate covariates to control for population stratification in the association analysis. LD-pruning was performed using PLINK, and only independent (r2 < 0.2), autosomal non-GC/AT SNPs were included in the PCA, which was conducted using EIGENSTRAT (Price et al., 2006) version 3.0.

Association analysis and meta-analysis

The single-marker analysis for the genome-wide data was carried out using logistic regression on allele counts with the first 10 principle components as covariates. P values and odds ratios with the corresponding 95% confidence intervals were calculated for the association analysis in Chechen and Circassian subpopulations separately. Both association and meta-analysis were performed using PLINK.

The WTCCC cohort

The cohort of European population was from WTCCC, which has been reported before (Wellcome Trust Case Control Consortium, 2007). All the samples were genotyped on Affymetrix Genome-Wide Human SNP Array 5.0. We similarly performed sample and SNP based QC steps and excluded non-European subjects based on PCA. Logistic regression was performed including the first three principal components as covariates.

Imputation analysis

The regional imputation at the locus of chr12q24.31 was conducted in two steps. First, the genotype data were prephased with SHAPEIT (Delaneau, Marchini & Zagury, 2012; Delaneau, Zagury & Marchini, 2013) version 2, and then genotype imputation was performed using IMPUTE 2 (Howie, Donnelly & Marchini, 2009; Marchini et al., 2007) with the 1000 Genome Phase 3 (https://mathgen.stats.ox.ac.uk/impute/1000GP%20Phase%203%20haplotypes%206%20October%202014.html) as the reference panel. Missing data likelihood score test was conducted to assess the association of each imputed SNP genotype with T2D using software SNPTEST (Marchini et al., 2007) V2, including the first three principal components as covariates. SNPs with info score <0.8 or with HWE-test p-value <1 × 10−06 were excluded from association testing.

Analysis of methylation data

Genomic DNA of a subset of samples in the biorepository of CAG was isolated from peripheral blood mononuclear cells. Genome-wide methylation profiling was conducted on the Infinium HumanMethylation450 BeadChip Kit at CAG according to the manufacturers’ protocols. Methylation data were exported from the Illumina GenomeStudio and loaded into the R statistical package (r-project.org) using the lumi package (Du et al., 2010; Lin et al., 2008). After adjusting for quantile color balance and background level and simple scaling normalization, M-value density and CpG-site intensity were plotted and aberrant chips were removed. These samples have also been genotyped at CAG and their genetic ethnicity was checked by PCA. We extracted the M-values (the log2 ratio between the methylated and unmethylated probe intensities) and the genotype information of the 425 subjects of European ancestry. We removed subjects of missing genotype at SNP rs4758690 and extreme outlier values of methylation M-values (≥median M-value of the genotype group ± 3SD) and then assessed the association between the additive genotype at rs4758690 and methylation M-value in gene MLXIP using linear regression including sex, age, and 10 genotype-derived principle components. Box-plots were generated using R package.

Results

Identification of novel T2D signals in Jordan subpopulations

To understand the genetic basis for T2D in Jordan populations, we conducted GWAS in Chechen and Circassian subpopulations of Jordan. The sample information after QC is summarized in Table 1. Specifically, for the Chechen subpopulation, we have 34 cases and 109 controls; for the Circassian subpopulation, we have 33 cases and 105 controls (Table 1). Approximately 645,000 SNPs in each subpopulation passed QC. We conducted logistic regression analyses separately in each population, including ten genotype-derived principal components as covariates. There was no signal that reached genome-wide significance, however there are several SNPs at suggestive level of significance (P < 1 × 10−4) in each subpopulation (Tables S1–S2). Then we performed meta-analysis of the association results from the two subpopulations. In the meta-analysis, we observed a signal at genome-wide significant level (SNP rs6134031, P-value = 1.12 × 10−8) under both fixed effect model and random effect model (Fig. S1, Fig. 1, Table 2). This SNP is located at the 5′ of the JAG1 gene (Fig. 1). In addition, there is another signal with multiple SNPs showing suggestive evidence of association (P-value <  1 × 10−4), with SNP rs4758690 having the lowest P-value at 4.20 × 10−5 (Fig. S1, Table 2, Fig. 1). SNP rs4758690 is located in the intron of MLXIP, a gene involved in transcriptional regulation of genes in glucose metabolism. Taken together, these results demonstrate significant GWAS associations to novel T2D susceptibility loci in Jordan subpopulations.

Table 1 The number of samples after quality control filtering.

Ethnicity	Cases	Controls	Total	
	N	Male %	N	Male %	N	
Chechen	34	47%	109	40%	143	
Circassian	33	39%	105	45%	138	
Total	67		214		281	
Notes.

N Number

Figure 1 The regional association plots for the top associated loci.

(A) chr20p12.2 locus in Circassian population; (B) chr20p12.2 locus in Chechen population; (C) chr12q24.31 in Chechen population. The top associated SNP at each locus is shown in purple and the LD between the remaining SNPs and the index SNP are indicated by their colors. The r2 values were calculated from the each population using software PLINK (Purcell et al., 2007). The recombination rates are shown by the light blue lines and the genomic positions are on human genome build hg19. The plots were made using software LocusZoom (Pruim et al., 2010).

Table 2 Top associations (P < 5 × 10−5) found in meta-analysis of Circassian and Chechen subpopulations.

SNP	Chr	Pos (hg19)	Gene	A1/A2	Ethnicity	MAF cases/controls	OR (95% CI)	P-value	
rs6134031	20	10752610	JAG1	T/C	Circassian	0.50/0.25	9.48 (3.09,29.07)	8.36 × 10−5	
					Chechen	0.51/0.23	9.84 (3.33,29.02)	3.45 × 10−5	
					Meta		9.66	1.12 × 10−8	
					European	0.28/0.26	1.12 (1.03,1.23)	0.012	
rs4758690	12	122610909	MLXIP	G/A	Circassian	0.59/0.41	2.41 (1.19,4.91)	0.015	
					Chechen	0.60/0.38	3.89 (1.78,8.47)	6.36 × 10−4	
					Meta		3.00	4.20 × 10−5	
					European	0.53/0.52	1.01 (0.93,1.09)	0.61	
Notes.

SNP single nucleotide polymorphism

Chr chromosome

Pos Position

A1 minor allele

A2 major allele

MAF minor allele frequency

OR odds ratio

CI confidence interval

Test the association signals in European population

We then investigated whether these association signals exist in populations of other ethnicities. We examined the association of these SNPs in the T2D dataset of the Wellcome Trust Case Control Consortium (WTCCC) (Wellcome Trust Case Control Consortium, 2007) which is composed of 1,999 cases and 3,004 controls, genotyped on the Affymetrix Genome-Wide Human SNP Array 5.0. After QC, 1,952 cases and 2,960 controls of European ancestry remained for association analysis by logistic regression. The top SNP in the Jordan analysis, rs6134031 demonstrated nominally significant association with T2D in the WTCCC cohort (P = 0.012) and the same direction of effect (Table 2). The SNP rs4758690 is not genotyped on the Affymetrix GW5.0 Array, so we conducted imputation over this region in the replication cohort. Based on the imputed genotype data, we did not observe a significant association to rs4758690 (OR = 1.01, P = 0.61).

Correlation of T2D variants with MLXIP gene expression and methylation

Interrogating these T2D variants in the GTEx dataset (GTEx Consortium, 2015), we uncovered a nominally significant association between SNP rs6134031 and JAG1 expression, in Esophagus–Muscularis (Beta = − 0.15, P = 0.0073, Fig. S2) and a marginal correlation in pancreatic tissue which is of potential biological relevance to T2D (Beta = −0.13, P = 0.071, Fig. S2). Though it is not significant, we did observe a trend of association between the doses of minor allele T and a lower expression of JAG1.

On the other hand, we found a genome-wide significant eQTL effect of SNP rs4758690 for gene MLXIP expression in transverse colon (Beta = 0.46, P = 1.10 × 10−14) and small intestine terminal ileum (Beta = 0.50, P = 4.20 × 10−7) tissue specimens (Fig. 2). A similar significant eQTL effect was reported for MLXIP expression in normal pre-pouch ileum in another study examining eQTLs in human intestine tissues (Kabakchiev & Silverberg, 2013).

Figure 2 Box plots showing the association between SNP rs4758690 genotype and gene MLXIP expression level.

(A) in tissue transverse colon, beta = 0.46, P = 1.10 × 10−14; (B) in tissue small intestine, beta = 0.50, P = 4.20 × 10−7. The in silico analyses were conducted at GTEx Protal (GTEx Consortium, 2015). The sample groups of different rs4758690 genotype were indicated on the X-axis; and the relative expression level of MLXIP is shown on the Y-axis. The median value of MLXIP expression level in each genotype group is represented by the dark black horizontal line in the box plot. In the both figures, the reference allele is G and the alternative allele is A.

Further, we found that SNP rs4758690 is significantly associated with the methylation probe cg22729539 (P = 3.07 × 10−5) residing within an intron of the longest isoform of MLXIP (Fig. 3). This site is absent in other short isoforms. We observed a positive correlation between the eQTL and the methylation data at this locus. As methylation is one of the important mechanisms regulating gene expression, these results are of potential interest. The minor allele G confers a lower expression of MLXIP compared to the major allele A, as well as a reduced methylation level at probe cg22729539, consistent with previous reports that gene body methylation was found to be positively correlated with gene expression (Yang et al., 2014). In addition, cg22729539 resides in a region with multiple histone modifications and transcription factor binding in pancreatic islets and liver cells which are central to T2D (Fig. S3) and additional T2D relevant cell lines (Table S3) (Bhandare et al., 2010; Encode Project Consortium, 2012; Parker et al., 2013; Pasquali et al., 2014; Roadmap Epigenomics et al., 2015). The bound transcription factors include CEBPB which is known to function in adipogenesis (Darlington, Ross & MacDougald, 1998), ER stress and pancreatic β cell failure (Matsuda et al., 2010) (Table  S3), therefore this region may function as active cis-regulatory element, regulating MLXIP expression.

Figure 3 The association between SNP rs4758690 genotype and methylation status in gene MLXIP.

M-values for methylation probe cg22729539 are plotted against the additive genotype at SNP rs4758690. Dark horizontal lines in the boxplots indicate the median M-value of each genotype group, the boxes represent the first to third quartiles, and the ends of whiskers of the boxplot show 1.5 times the interquartile range (IQR). Open circles represent data points outside of the range of 1.5 IQR. Red diamonds indicate the means of each genotype group, with the values of the mean ± standard deviation shown in red text. The number of individuals in each group with additive genotype of minor allele G is shown below the X-axis.

The expression of JAG1 and MLXIP

The biological relevance of these two genes to T2D was further strengthened by their expression pattern. For JAG1, it is reported to be highly expressed in arteries and in bronchial epithelial cells and lung tissue, with a particularly high level of expression in the gastrointestinal tract tissues, such as small intestine and colon (Figs. S4 and S5). For the gene MLXIP, high levels of expression have been consistently noticed in colon tissue as reported in different studies (Figs. S6 and S7). Both of these genes demonstrated medium level of expression in certain tissues highly relevant to T2D, including JAG1 in adipose, pancreas, and smooth muscle (Figs. S4 and S5), and MLXIP in muscle, pancreas and pancreatic islet cells (Figs. S6 and S7).

The overall expression pattern of JAG1 is similar to that of the gene Coagulation Factor III (F3) (correlation >  0.7), genetic polymorphisms of which have been shown to be associated with T2D in different ethnicity groups (Palmer et al., 2012; Yamada et al., 2006; Yamaguchi et al., 2007) and the expression of which is significantly higher in monocytes and neutrophils of diabetes and prediabetic subjects (Ichikawa et al., 1998).

Consistent with the expression pattern, knockout of JAG1 in a mouse model resulted in defects in endocrine/exocrine glands, homeostasis/metabolism, and the liver/biliary system (Fig. S8) (Blake et al., 2017; Finger et al., 2017). MLXIP-deficient mice displayed distinct metabolic features including increased serum lactate and alanine levels, consumption of fatty acids for energy production during exercise, and increased glycolytic capacity in skeletal muscles. These features are associated with T2D in humans (Crawford et al., 2010; Imamura et al., 2014; Karpe, Dickmann & Frayn, 2011).

Replication of previously reported T2D loci

Previous genetic and genomic studies of T2D have yielded fruitful results. Based on literature review and a search of the NHGRI-EBI GWAS catalog (Welter et al., 2014), we generated a list of 182 genes which have been reported to be associated with T2D. Among them, 86 have intragenic SNPs or nearby SNPs that are nominally significant in our meta-analysis of Jordan subpopulations (Table S4), demonstrating the validity of our study even with a small sample size and support for common genetic basis of T2D in different ethnicities.

Discussion

In this first GWAS of T2D in Jordan subpopulations, we identified a novel genome-wide significant locus at chr20p12.2 close to gene JAG1 and replicated the association in the samples of European ancestry of the WTCCC dataset. JAG1 is expressed in T2D relevant tissues and knockout of JAG1 resulted in T2D related phenotypes in mice. We also found an interesting locus of suggestive significance at 12q24.31 in the intron of MLXIP. We further showed there is strong eQTL effect of the top associated SNP at this locus with correlation between its genotype and methylation of MLXIP, suggesting this locus may confer a cis-regulatory effect on MLXIP expression and this effect is at least in part mediated through methylation.

JAG1 encodes a ligand for receptor Notch 1, functioning in the Notch signaling pathway which is important for multiple cellular functions, especially during normal development and pathogenesis of cancer (Bray, 2016). Accumulative evidence demonstrate a critical role of the Notch signaling pathway in the regulation of metabolism and that perturbations in Notch signaling may lead to the development of obesity and T2D. It has been shown that overactivation of Notch signaling results in stimulation of glycogenolysis and gluconeogenesis in the liver, counteracting insulin effects (Bi & Kuang, 2015; Pajvani et al., 2013; Pajvani et al., 2011). Another role of Notch signaling in diabetes mellitus is to increase lipogenesis via mechanistic target of rapamycin complex 1, resulting in the development of hyperglycemia and fatty liver (Bi & Kuang, 2015; Pajvani et al., 2013), dysfunctions associated with T2D. Positive correlation of Notch signaling with insulin resistance and fatty liver has been reported in humans (Valenti et al., 2013). Key roles of Notch signaling also include regulation of adipocyte homeostasis and skeletal muscle homeostasis (Bi & Kuang, 2015). One upstream regulator of JAG1, HMGA1 is also involved in the molecular mechanism of T2D (Bianco et al., 2015). It has been reported that the expression of JAG1 is down-regulated upon HMGA1 depletion by siRNA (Pegoraro et al., 2013). HMGA1 encodes a non-histone chromatin associated protein, involved in multiple important cellular functions underlying pathogenesis of T2D, such as insulin production (Arcidiacono et al., 2014), in insulin action (Iiritano et al., 2012).

MLXIP encodes MondoA which interacts with MLX. Together they activate transcription of genes involved in glucose metabolism (Sloan & Ayer, 2010). Recent studies demonstrate that in addition to regulation of glucose-sensing transcription, MLXIP plays an important role in Myc activation and subsequent metabolic pathway reprogramming (Carroll et al., 2015). It is well known that Myc has important functions in the pathogenesis of diabetes, through both regulating cell cycle entry and maintaining expansion, regeneration and normal function of beta-cells (Tiwari et al., 2016). It has been shown that abnormal activation of Myc resulted in decreased beta-cell differentiation, proliferation and reduced insulin secretion (Cheung et al., 2010). On the other hand, insufficient Myc expression leads to hyperglycemia and beta-cell inactivity (Guo et al., 2013).

The pathological events that can lead to the development of T2D are diverse, such as deficiency and malfunction of beta-cells together with insulin resistance in multiple tissues, including liver and adipose tissues (Tiwari et al., 2016). The likely underlying genes for the novel T2D signals that we identified through GWAS are key players of signaling pathways that could lead to the development of T2D.

It is interesting that in our study, we observed a positive correlation between methylation and MLXIP expression that was associated with the rs4758690 SNP. While methylation at promoter sites usually results in gene silencing, methylation at other gene sites often enhances gene expression (Yang et al., 2014) or affects splicing (Jones, 2012). The presence of histone modification marks and transcription factor binding in the vicinity of methylation probe cg22729539 suggests that this region contains cis-regulatory elements that actively regulate transcription. These epigenetic factors, like DNA methylation and histone modification, may interact with each other to influence gene expression in either the same or opposite directions (Banovich et al., 2014; Cedar & Bergman, 2009). DNA methylation could also affect nearby transcription factor binding, such as transcription factor CEBPB, which plays an important role in adipogenesis (Darlington, Ross & MacDougald, 1998), ER stress and pancreatic β cell failure (Matsuda et al., 2010). The coordination between a variety of genetic and epigenetic factors may regulate the expression of MLXIP, and further the development of T2D.

The two SNPs, rs6134031 and rs4758690 have been reported to be associated with other human traits, though genome-wide significance was not reached in those studies. In the NHGRI-EBI GWAS catalog (Welter et al., 2014), SNP rs6134031 has been reported to be associated with Plasma omega-6 polyunsaturated fatty acid levels (linoleic acid, n-6 PUFAs ) (rs6134031-T, beta = 0.0372, P-value = 4 × 10−6) (Dorajoo et al., 2015). The relationship between n-6 PUFAs and T2D is debatable. Generally, n-6 PUFAs are considered to be proinflammatory and n-3 PUFAs to be anti-inflammatory. Thus, high dietary intake of n-6 PUFAs and elevated (n-6) to (n-3) ratio are associated with chronic inflammatory diseases including T2D (Patterson et al., 2012; Simopoulos, 2016). However, a recent study by Forouhi et al. (2016) in a large number of European subjects found that different types of n-6 PAFUs are differentially associated with risk of T2D. Linoleic acid (LA) and eicosadienoic acid (EDA) were inversely associated with T2D (OR < 1), arachidonic acid (AA) was not significantly associated, and γ-linolenic acid (GLA), dihomo-GLA, docosatetraenoic acid (DTA), docosapentaenoic acid (n6-DPA) are positively associated (OR >  1). Thus the relationship between n-6 PUFAs (and its subtypes) and T2D needs to be further evaluated in more studies. SNP rs4758690 is also associated with height (P-value = 2.396 × 10−5), however the effect size and direction of effect are not available (Lango Allen et al., 2010). A systematic review and meta-analysis of 18 studies revealed that significant inverse association between height and T2D risk was only observed in women, not men (Janghorbani, Momeni & Dehghani, 2012). Thus the genotype of these 2 SNPs are important for inter-related human traits, suggesting these traits share common molecular underpinnings.

Our study has started to reveal the similarities and differences of the genetic basis of T2D between Jordan subpopulations and other ethnicities. Despite the small sample size, we were able to replicate almost half of the loci that were reported to be associated with T2D in genetic and genomic studies in other populations. The replication of these associations suggests some common genetic basis underlying the development of T2D among different ethnicities. For complex traits and diseases, there are many GWAS loci which could not be replicated across different ethnicities, such as the SNP rs7756992 in the CDKAL1 gene which strongly associates with T2D in subjects of European ancestry, but displayed no association in a population of West Africa (Steinthorsdottir et al., 2007). Among the 37 SNPs associated with T2D in European or Asian populations, only two were replicated in a Qatari population (O’Beirne et al., 2016). In the Jordan subpopulations examined, we observed a significant association of rs6134031 and T2D, with a very large effect size. In the WTCCC, including only subjects of European ancestry, the LD structure for this region is different and the association of rs6134031 with T2D is less strong. The association at SNP rs4758690 is nominally significant in both Jordan subpopulations, however it is not significant in WTCCC subjects of European ancestry. The identification of these two loci suggested unique genetic determinants for T2D in the Jordan subpopulations. The separate GWAS performed in Chechen and Circassian subpopulations also suggest distinct genetic factors for T2D in each of these two ethnicities. As reviewed by Rosenberg et al. (2010), such ethnic population differences may arise from variations in disease allele frequency, effect direction, effect size, distinct LD patterns, and trait/disease phenotype prevalence. Therefore, it is important to carry out genetic studies in different ethnic groups.

A major limitation of our study is the small size, which reduces the statistical power to detect a true effect of the genetic variants. The small sample size may lead to p-values of true associations failing to reach stringent significance thresholds, like the genome-wide significance threshold of 5 × 10−8, resulting in false negatives (type II error). Therefore, we also considered other biological evidence when interpreting our results and we were encouraged by the replication of the JAG1 locus and the strong eQTL signal observed for MLXIP, due to their strong biological relevance to T2D. As reported and discussed by other studies, true association may not always reach the conventionally corrected conservative threshold of 5 × 10−8 for declaring a genome-wide significance (Nishizawa et al., 2014). In our case, future studies with larger sample sizes of Jordan populations are needed to replicate the findings from our study and to further identify other genetic loci.

Conclusion

Taken together, our results from the first GWAS of T2D conducted in two subpopulations in Jordan have identified novel genetic factors underlying T2D; we additionally demonstrate there is common genetic basis among the different ethnicities as well as certain unique genetic factors that underlie T2D in the Jordan subpopulations. Identification of these novel genetic risk factors will offer the potential to gain further insight into the development of T2D and may help with the development of novel treatments precisely for the Jordan populations, which will reduce disease burden and promote health.

Supplemental Information

Supplemental Information 1 Supplementary materials

Click here for additional data file.

Supplemental Information 2 Raw data of phenotype and SNP genotype

Click here for additional data file.

We would like to thank the Circassian and Chechen communities for their cooperation in conducting this study.

Additional Information and Declarations

Competing Interests

Author Contributions

Human Ethics

Data Availability

The authors declare there are no competing interests.

Rana Dajani conceived and designed the experiments, contributed reagents/materials/analysis tools, wrote the paper, reviewed drafts of the paper.

Jin Li, Zhi Wei and Michael E. March performed the experiments, analyzed the data, wrote the paper, prepared figures and/or tables, reviewed drafts of the paper.

Qianghua Xia analyzed the data, prepared figures and/or tables, reviewed drafts of the paper.

Yousef Khader, Nancy Hakooz, Raja Fatahallah, Mohammed El-Khateeb, Ala Arafat, Tareq Saleh, Abdel Rahman Dajani, Zaid Al-Abbadi, Mohamed Abdul Qader, Abdel Halim Shiyab, Anwar Bateiha and Kamel Ajlouni contributed reagents/materials/analysis tools, reviewed drafts of the paper.

Hakon Hakonarson conceived and designed the experiments, contributed reagents/materials/analysis tools, wrote the paper, reviewed drafts of the paper.

The following information was supplied relating to ethical approvals (i.e., approving body and any reference numbers):

The study has been approved by the Institutional Review Board committee at the National Center for Diabetes, Endocrinology and Genetics of Jordan. The written informed consent was given by all participants.

The following information was supplied regarding data availability:

The raw data has been uploaded as a Supplementary File.

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
