# Peer review of "Genome-wide association study identifies novel type II diabetes risk loci in Jordan subpopulations"

_PeerJ, doi:10.7717/peerj.3618_

## Round 0.1 · original submission · Major Revisions

Reviewer 1 focuses mainly on statistical issues, and reviewer 2 on JAG1 functional aspects. Please, do not neglect grammar mistakes/typo errors.

Reviewer 1 ·

Basic reporting

No comment.

Experimental design

No comment.

Validity of the findings

No comment.

Additional comments

In this paper, the authors performed a GWA study to identify gene polymorphisms associated with T2DM in Circassian and Chechen ethnic groups. Further, authors replicated their study in a Caucasian population and performed a meta-analysis of overall findings. The paper is susceptible for publication, but I have some comments that need to be addressed by the authors:

Major concerns
1) The strong limitation of the present study is the very low sample size. This limitation does not allow to exclude a β error when a significant result is not reached. The authors must discuss this limitation.
2) Have the continuous variables a normal distribution? If not, was a log-transformation performed to approximate a normal distribution before the linear regression?
3) Authors must precise if the “fixed” or “random effect” was used in meta-analysis.
4) JAG1 is regulated by HMGA1 (Pegoraro et al. Oncotarget 2013), an architectural transcription factor involved in insulin production (Arcidiacono et al. Front Endocrinol. 5:237; 2015), in insulin action (Iiritano et al. Mol Endocrinol. 26(9):1578-89; 2012) and in T2DM (Bianco et al. PLoS One. 10(8):e0136077; 2015). Please, discuss this issue in Discussion section and cite previous papers.

Minor points
1) There are some typo errors in the main text.

Reviewer 2 ·

Basic reporting

The format of Figures and Tables should be improved: For example,
Figure 1 legend: Line 594: ‘Figure 1 (a): The regional association ….’ Should be ‘Figure 1 : …..’;
Figure 1: the r2 box in (c) is not at the same position as (a)(b)

The table should change its current format to a more professional one, including the lines, fonts, and position of contents in the table (e.g. align left/right).

In both text body and tables, the P-values should be rounded to a consistent digit number. And the P-val format (e.g 1e-14 or 1×10-14) should also follow the journal’s requirements throughout the manuscript.

Line 286: Here should contains the citation of NHGRI-EBI GWAS catalog.

Experimental design

no comment

Validity of the findings

The authors did not find a significant association between rs6134031 and JAG1 expression, is there any other possible mechanisms for this rs6134031, e.g. affecting histone modification, act as a trans-eGene? Or could it be LD-related SNPs? In addition, I suggest authors to double check the gene ID in supplementary Figure 2 since ENSG00000270408 for JAG1 has been deprecated.

I suggest the authors show the results of PCA. It is an important indicator to show ethnic identity and population control.

The authors checked the knockout results of JAG1 in a mouse model. How about the results of MLXIP?

Have the authors checked the if these two SNPs are reported in other diseases or phenotypes? If so, is there any relation between these traits and T2D?

Line 278-280: The authors argued JAG1 and MLXIP expression showed high expression in T2D relevant tissues, however, when I checked the Supplementary Figure 3-6, I found this conclusion is not so validated. For example, the expression of JAG1 in muscles in Supplementary Figure 3 ranked very low. Also, please organize the data from BioGPS based on the tissues and show the expression results from highest to lowest.

Additional comments

Line 106: The authors there has been no GWAS of T2D specifically conducted among Middle-East populations, however, I found one study by Michella Ghassibe-Sabbagh et al (doi:10.1038/srep07351), in which they performed a T2D GWAS in the Lebanese population. I am not familiar with the difference among populations in Mid-East, thus, I suggest the authors provide a reasonable explanation here or use a more accurate expression to describe the novelty of their study.

Line 266-268: Authors stated that ‘cg22729539 resides in a region with multiple histone modifications in T2D relevant cell lines’, I suggest they better use a more straightforward presentation, i.e figures of histone modification and DNaseI using UCSC genome browser.

Line 230: I think here ‘These SNPs’ may be a typo. It should be rs4758690.

Line 242: the results are in table 2, not table 1.

The manuscript still has many problems in gramma and format, eg.
Line 225: ‘Supplementary Table 1-2’ should be ‘Supplementary Tables 1-2’;
Line 245: ‘rs4758690; OR= 1.01, P=0.61’ should be like ‘rs4758690(OR= 1.01, P=0.61)’

---

## Round 0.2 · accepted · Accept

The authors have satisfactorily answered to the reviewers' concerns.

Reviewer 1 ·

Basic reporting

No comment

Experimental design

No comment

Validity of the findings

No comment

Reviewer 2 ·

Basic reporting

In the revised version, the authors appropriately responded to the previous comments

Experimental design

In the revised version, the authors appropriately responded to the previous comments

Validity of the findings

In the revised version, the authors appropriately responded to the previous comments

Additional comments

In the revised version, the authors appropriately responded to the previous comments